# Mating Disruption of *Helicoverpa armigera* (Lepidoptera: Noctuidae) on Processing Tomato: First Applications in Northern Italy

**DOI:** 10.3390/insects11040206

**Published:** 2020-03-26

**Authors:** Giovanni Burgio, Fabio Ravaglia, Stefano Maini, Giovanni Giorgio Bazzocchi, Antonio Masetti, Alberto Lanzoni

**Affiliations:** Department of Agricultural and Food Sciences, Alma Mater Studiorum-Università di Bologna, Viale G. Fanin, 42, 40127 Bologna, Italy; fabio.ravaglia@studio.unibo.it (F.R.); stefano.maini@unibo.it (S.M.); giovanni.bazzocchi@unibo.it (G.G.B.); antonio.masetti@unibo.it (A.M.); alberto.lanzoni2@unibo.it (A.L.)

**Keywords:** mating disruption, cotton bollworm, processing tomato, geostatistics

## Abstract

*Helicoverpa armigera* is a polyphagous and globally distributed pest. In Italy, this species causes severe damage on processing tomato. We compared the efficacy of mating disruption with a standard integrated pest management strategy (IPM) in a two-year experiment carried out in Northern Italy. Mating disruption registered a very high suppression of male captures (>95%) in both growing seasons. Geostatistical analysis of trap catches was shown to be a useful tool to estimate the efficacy of the technique through representation of the spatial pattern of captures. Lower fruit damage was recorded in mating disruption than in the untreated control plots, with a variable efficacy depending on season and sampling date. Mating disruption showed a higher efficacy than standard IPM in controlling *H. armigera* infestation in the second season experiment. Mating disruption showed the potential to optimize the *H. armigera* control. Geostatistical maps were suitable to draw the pheromone drift out of the pheromone-treated area in order to evaluate the efficacy of the technique and to detect the weak points in a pheromone treated field. Mating disruption and standard IPM against *H. armigera* were demonstrated to be only partially effective in comparison with the untreated plots because both strategies were not able to fully avoid fruit damage.

## 1. Introduction

*Helicoverpa armigera* (Hubner) (Lepidoptera: Noctuidae), the African bollworm or cotton bollworm, is distributed worldwide with the exception of North America. This polyphagous pest causes severe damage to many crops including tomato, cotton, pea, chickpea, sorghum, and cowpea [1]. The severity of cotton bollworm damage varies by crop and region and is influenced by the temporal scale [2]. Due to its dispersive and migratory behavior, the incidence of this pest is often unpredictable. In Italy, the severity of damage caused by *H. armigera* has increased in recent years [3], especially on processing tomato. *Helicoverpa armigera* is listed as a quarantine pest by the European and Mediterranean Plant Protection Organization [4,5].

Italy is the most important tomato (*Licopersicon exculentum* Mill) producer in Europe, and the Emilia-Romagna region (Northern Italy) accounts for approximately 30% of all Italian production [6]. Emilia-Romagna is also the leader in organic tomato cultivation (69% of Italian production) because of favorable pedoclimatic conditions and the presence of important processing companies. The area of organic tomato production is steadily increasing due to the farmers’ profit and the high demand for organic tomatoes by consumers.

A high level of resistance to chemical insecticides by *H. armigera* (i.e., carbamates) has led to control failures in many parts of the world including Europe [7]. For this reason, there is an increasing interest in alternative approaches to controlling this pest. Pheromones have been utilized in a variety of ways including mass trapping, disruption of mating communication, monitoring, and surveying [8,9]. Mating disruption has been tested on the Noctuidae species including *Spodoptera* spp. infesting vegetable crops, onions, cotton, herbs, and ornamentals, both in open fields and greenhouses [10,11,12,13,14,15,16], and has been shown to be an effective method to control pest populations. However, even if pheromone application has been found to disrupt the males’ ability to locate a pheromone source, in some cases, the larval populations were not reduced to the point at which insecticide sprays could be eliminated [14]. Mating disruption for *H. armigera* management has been demonstrated to be effective in reducing field infestations [17,18,19]. However, with this species, the efficacy of mating disruption also showed a certain degree of variability and in some cases, a lower level of control was achieved in comparison to the chemical sprays. A number of factors including field size, crop species, receiving environment, local climatic characteristics (e.g., dominant winds), behaviors of insect target species (e.g., pre-oviposition flight behavior of mated female [19]), can influence the field efficacy of this technique.

Mating disruption seems to be particularly suitable for processing tomato in Northern Italy where this crop is cultivated on wide areas characterized by an intensive cropping system. Moreover, this technique is characterized by the lack of negative impacts and can be integrated with reduced risk insecticides like microbial products.

The objective of the study was to evaluate the efficacy and feasibility of mating disruption to control *H. armigera* infestations in an area where processing tomato are intensively grown by analyzing both the reduced trap capture and subsequent fruit damage reduction. The efficacy of this technique was also compared with a standard integrated pest management strategy (IPM) by means of fruit damage evaluation [20]. The field trials were carried out on a farm representative of tomato cultivation in the Emilia-Romagna region.

## 2. Materials and Methods

### 2.1. Site Description

The study was conducted in an area cropped with processing tomato located in Ravenna Province, the Emilia-Romagna region, Italy. A pilot farm, managed using integrated pest management (IPM) methods and representative of the tomato cultivation conditions of the region, was selected. The IPM method consisted of a strategy according to which insecticide need was determined on the basis of insect density monitoring performed twice a week.

### 2.2. Mating Disruption Trials Planning

In each year, two treatments were compared: (i) mating disruption (i.e., pheromone-treated) and (ii) the control (i.e., pheromone-untreated), where pheromones were not applied. The field experiments were carried out in two consecutive growing seasons, 2011 and 2012, in two nearby fields. In both years, insecticide sprays were applied across the whole tomato field, according to the IPM strategy (Table 1). In 2011, a 1-ha pheromone-treated area and a 1-ha pheromone-untreated control area (both approximately 100 × 100 m) were delimited into the western half of an 18-ha tomato field (44°30’31”N, 12°11’43”E). In 2012, two areas (5 ha each, both approximately 180 × 270 m) were delimited within a 15-ha tomato field (44°30’12”N, 12°11’13”E), and designated pheromone-treated and pheromone-untreated (control). In each year, the pheromone untreated control area was located upwind, approximately 300 m away from the pheromone-treated area to avoid pheromone drift into the pheromone-untreated control area. Pheromone-treated and pheromone-untreated control areas were divided into four quadrants for replication purposes. 

### 2.3. Pheromone Treatments 

BioSelibate HA dispensers (Suterra Europe, Valencia, Spain), consisting of a sawdust type material each containing 0.29 g a.i. (Z-11 hexadecenal and Z-9 hexadecenal in a ratio of 91:9), were used. The target application rate was 100 dispensers/ha (=29 g a.i./ha) that were manually hung on 0.8 m high rods in a 10 × 10 m grid. 

In 2011, the experiment started on May 27 (dispensers were hung on May 27; tomato was transplanted on May 25 and 26) and continued until August 16. The tomato was harvested from August 19 to 22. In 2012, the experiment started on June 20 (the dispensers were hung on June 20; tomato was transplanted from June 6 to 8) and tomato was harvested on September 7. Currently, the pheromones for the mating disruption of *H. armigera* are not yet commercial and were provided by Suterra to cover a total of 6 ha. 

### 2.4. Male Capture Evaluation

Pheromone-baited traps (AgriSense funnel trap green/yellow/transparent) were used to verify if male cotton bollworm moths were able to locate a pheromone source in the pheromone-treated area. Trap catches in the pheromone-treated area were compared with those recorded in the pheromone-untreated control area. Four traps in the 2011 trial and eight in the 2012 trial were placed in each of the pheromone-treated and untreated control areas. Traps were baited with *H. armigera* pheromone lures (Septa pheromone lure, Suterra). Baited traps were hung at about 80–100 cm above the ground and at least 10 m inside the area and 35 m far apart from each other. Each trap had an insect killing strip (a.i., 15% diazinon) at the bottom of the trap. Male moths were collected from the traps weekly. In addition to these traps, a grid of pheromone baited traps (26 traps in total in 2011 and 42 traps in 2012) were also placed to cover all the area of the tomato field where the areas were delimited with the aim to map the area where mating disruption may have been effective. All the traps were georeferenced using a handheld Magellan SporTrak Map^®^ GPS unit.

### 2.5. Fruit Damage Estimation

Within the pheromone-treated and pheromone-untreated control areas, four plots left without insecticide sprays (one per quadrant) were delimited with the aim of sampling for the evaluation of fruit damage. Likewise, four plots were also selected within an area of the field receiving only the chemical spray to control *H. armigera*.

Cotton bollworm damage on tomato (proportion of damaged fruits) in the pheromone-treated, pheromone-untreated control, and insecticide treated areas was estimated by visual inspection using a sample of fruits from within each of four plots nested within each treatment, over six consecutive samplings. In both years, samplings 1–6 corresponded to weeks 7–12 after transplanting (WAT). In each of the four plots, the samplings were performed by checking the fruits for 30 s on 10 randomly selected plants. Fruit damage estimation in each treatment was taken on a weekly basis. On each sampling date, a different set of plants were sampled. Fruit damage estimation followed a stratified design, with treatments (n = 3) nested into years (n = 2), plots (n = 4) nested into each treatment, and sampling dates (n = 6) in each year.

### 2.6. Data Analysis

#### 2.6.1. Male Capture Analysis

The comparison of the male catches in the pheromone-treated and pheromone-untreated areas was analyzed by the Mann–Whitney U test (*p* < 0.05). The ratio of the catch reduction in the pheromone-treated area with respect to the pheromone-untreated area was calculated as follows:
$$\mathrm{Catch}\;\mathrm{reduction}\;\mathrm{ratio}=\frac{\mathrm{Pheromone}\;\mathrm{untreated}\;\mathrm{area}\;\mathrm{catch}-\mathrm{pheromone}\;\mathrm{treated}\;\mathrm{area}\;\mathrm{catch}}{\mathrm{Pheromone}\;\mathrm{untreated}\;\mathrm{area}\;\mathrm{catch}}\times 100$$

Data from the pheromone traps were also analyzed using geostatistics, in order to compare the spatial pattern of the male captures between the pheromone-treated (mating disruption) area and the pheromone-untreated (control) area. One of the main objectives of geostatistical studies is to provide a spatial representation of data by estimating variable values at unsampled locations. Geostatistics offers a great variety of interpolation methods including stochastic techniques like kriging, and deterministic methods like inverse distance weighting (IDW) [21,22]. IDW was selected as the interpolation tool to provide a visual representation of the population pattern in pheromone-treated and pheromone-untreated areas. Maps estimated by IDW were validated by cross-validation analysis, comparing the predicted and observed trap catch values using linear correlation analysis [23]; in addition, the mean prediction errors of the estimates were calculated. Geostatistical analysis was employed using ArcGIS, with the geostatistics ARCMAP extension (ESRI, Redlands, CA, USA).

#### 2.6.2. Fruit Damage Assessment

In each sampling date, fruit damage was calculated as the ratio of damaged fruits on the total of fruit sampled; standard errors of the damage ratio were calculated according to a binomial distribution [24]. The ratio of damaged fruits was analyzed using log linear analysis, a method that mimics a factorial analysis of variance [25] and allows for simultaneous evaluation of multiple interactions among categorical variables. Log linear analysis uses a likelihood ratio statistic *χ*² that has an approximate *χ*^2^ distribution. In our log linear analysis, the response variable was the ratio of damaged fruits, while the design (or independent) variables were: years (2011–12), treatments (mating disruption–chemical–untreated control), and sampling dates (n = 6). Additionally, a model involving “plot” (n = 4) as the design variable was tested, but this variable was removed because it did not show significant interaction with the response variable and the other design variable (*p* > 0.05). Although all interactions between variables were calculated by log linear analysis, only associations of the response variable (proportion of fruit damage) with design variables were taken into account for data interpretation.

The fruit damage recorded in the twelfth week after transplant (WAT 12) was the most relevant for the final evaluation of the treatment efficacy because it was the last sampling date before the harvest. For this reason, the frequency of damaged fruit at WAT 12 was analyzed by the *χ*^2^ test followed by a z-test to compare the column proportions and rank the efficacy of the treatments [26]. Bonferroni correction was implemented to adjust the *p*-level of the z-test. This procedure was performed, separately for each year, using the IBM SPSS 23 statistics package (IBM corporation, Armonk, NY, USA). 

## 3. Results

### 3.1. Mating Disruption Evaluation

The suppression of male captures was very high in both growing seasons (Figure 1, Table 2). In particular, the suppression ratio calculated for total capture was 99.2% and 98.4% in 2011 and 2012, respectively (Table 2). It is remarkable that capture suppression was also higher than 97% on the last sampling date at the end of August, corresponding to the harvest (Table 2).

The validation analysis of IDW maps is reported in Table 3 including the correlation analysis of the predicted vs. observed values and the calculation of the mean prediction errors. Eleven out of 12 of the contour maps were statistically supported by cross-validation analysis. In Figure 2 and Figure 3, the IDW maps of *H. armigera* male distribution, calculated from the total catch per trap during the sampling period, are shown. Only the maps of the total catch per year are reported, because they properly describe the spatial pattern of the catches during both full field seasons. In each year, the male catch within the pheromone-treated area can be visualized and compared with those of the remaining part of the pheromone-untreated field. The gaps of catches within the treated areas in the maps can be considered as a demonstration of the effectiveness of the male capture reduction due to mating disruption (Figure 2 and Figure 3). The catch patches, visualized as the darker filled contours, indicate the areas of the fields where male disruption was less or not effective. These areas of reduced efficacy correspond to the hedges and to the east zone of the mating disruption area, which is adjacent to the pheromone-untreated tomato. Moreover, the catch patches reached the highest values in the northeast area of the maps (up to 150 and 350 male captures in the 2011 and 2012 maps, respectively), corresponding to the pheromone untreated (control) area.

### 3.2. Fruit Damage Estimation

Lower fruit damage was found in the mating disruption and chemical spray areas than in the control plots for most sampling dates (Figure 4). No damage by other pests was detected during the field trials (e.g., *Tuta absoluta* (Meyrick) (Lepidoptera Gelechiidae). Data analysis showed single and multiple significant interactions between fruit damage and the design variables (Table 4) fruit damage between treatments (*p* < 0.001), years (*p* < 0.001), and sampling dates (*p* < 0.001). Overall, the damage on fruits was higher during 2012 (7.72%) than in 2011 (4.8%). Fruit damage had significant multiple interactions between variables and, for this reason, data were split in each field season and sampling date to provide an interpretation of the seasonal trend of fruit damage in each treatment.

It is remarkable that the relative efficacy of both control methods varied during the year, and this evidence was corroborated by the significant multiple interaction among “treatments * years * fruit damage” (Table 4); in particular, chemicals were more effective in 2011 than in 2012. Fruit damage was shown to also be dependent by treatments * sampling date and by treatment * year. Within each year and corresponding to the harvest (WAT 12), a *χ*^2^ test followed by a z-test was used to rank the efficacy of the treatments. Using this method, chemicals were scored as more effective than mating disruption in 2011 (Figure 4A), resulting in the following rank of efficacy: chemical > mating disruption > untreated control. On the other hand, chemical control was less effective in WAT 12 in the 2012 season, resulting in non-significant differences in comparison with the untreated control (Figure 4B). 

## 4. Discussion

Mating disruption demonstrated a variable efficacy in controlling *H. armigera*, measured by the analysis of fruit damage. A significant difference between control and mating disruption was obtained in both seasons, thus showing a robustness of the data obtained in the two-year replication of this study. In the 2011 season, the efficacy of mating disruption was lower than the chemical control, but in 2012, this trend was reversed. Overall, both control techniques against *H. armigera* were demonstrated to be only partially effective in comparison with the untreated control because the strategies were not able to fully avoid fruit damage. It is remarkable that mating disruption was more effective in the 2012 season when applied on a 5 ha field; in contrast, mating disruption applied on 1 ha (2011 season) resulted in a lower reduction of fruit damage. A wide area approach is a cornerstone of the mating disruption approach [9] and it could be hypothesized that the increased area of application in 2012 led to a higher efficacy of mating disruption.

The geostatistical analysis of trap catch reported in this study was a useful tool to evaluate the efficacy of mating disruption through the representation of the spatial pattern of catches. Catch gaps and patches can be interpreted as areas where catch reduction is optimal or ineffective, respectively. In particular, maps were used to verify how catch reduction was affected by the position of field dispensers and by dominant winds, in order to highlight and interpret potential border effects. The maps seem suitable to visualize the effects on male catches as a result of potential pheromone drift out of the pheromone-treated area, showing a partial efficacy of the capture reduction, in the downwind borders of the pheromone-treated area. Geostatistical techniques have been used to characterize the spatial and temporal variability of male *H. armigera* catches in Spain, affecting pest management actions, and as a powerful tool in precision agriculture systems [27]. That study demonstrated that moths were aggregated at the borders of tomato field, gradually colonizing the inner area on cloudless days when northeastern winds were predominant. A geostatistical analysis of the spatial heterogeneity of bollworm eggs was studied in China using semi-variance and kriging interpolation, providing a population risk analysis [21,28]. Authors showed that there was a high risk at early pest population stages (mid-June). Geostatistical maps of the spatial distribution of male catches proved to be suitable to analyze the efficacy of *Spodoptera littoralis* (Boisduval) (Lepidoptera Noctuidae) mating disruption [16] and to study the spatial distribution of vegetable pests including *Phthorimaea operculella* (Zeller) (Lepidoptera Gelechiidae) and the western corn rootworm *Diabrotica virgifera virgifera* LeConte, in Northern Italy [29,30]. Moreover, georeferencing tools can be used to decide on the best installation site according to topography and wind direction, when pheromone aerosol devices for mating disruption are intended to be employed [20]. A number of practical applications of spatial analysis in managing pests are reviewed and discussed by Sciarretta and Trematerra [22].

The reason for the low efficacy of the chemical treatment in 2012 is unknown, but this fact is in agreement with the field data of the extension services of the Emilia-Romagna region [31]. In particular, our data seem to corroborate the anecdotal findings that chemical control of *H. armigera* in Northern Italy tomato often achieves a partial and variable efficacy. In this context, the application of mating disruption in wide areas including adjacent fields treated with pheromones, seems to have the potential to enhance the efficacy of the technique. The crucial role of the pattern of dispenser distribution over a much greater area outside the cultivated field has been stressed by Kerns [14] and de Souza et al. [11], in order to make mating disruption effective. For these reasons, future application of the mating disruption technique should consider an increase in the number of dispensers in the areas adjacent to the hedges of the field in order to balance the reduced efficacy due to border effects [20,27]. 

## 5. Conclusions

Mating disruption could be applied as an IPM strategy to optimize *H. armigera* control in order to increase the efficacy of chemical or microbial control or to reduce the number of chemical sprays in highly infested areas. Pheromone dispensers for mating disruption of *H. armigera* are not commercially available yet, but we hope that this technique will be available soon for stakeholders involved in IPM and organic insect management. Further studies should take into account the application of mating disruption in combination with microbial agents like *Bacillus thuringiensis* subsp. *kurstaki*/*aizawai* and nuclear polyhedrosis virus (HaNPV), in order to promote a more ecological and sustainable control strategy to minimize the negative side effects of chemical control including the selection of insecticide-resistant strains and the harmful impact on beneficial insects occurring in agroecosystems.

## Figures and Tables

**Figure 1 insects-11-00206-f001:**
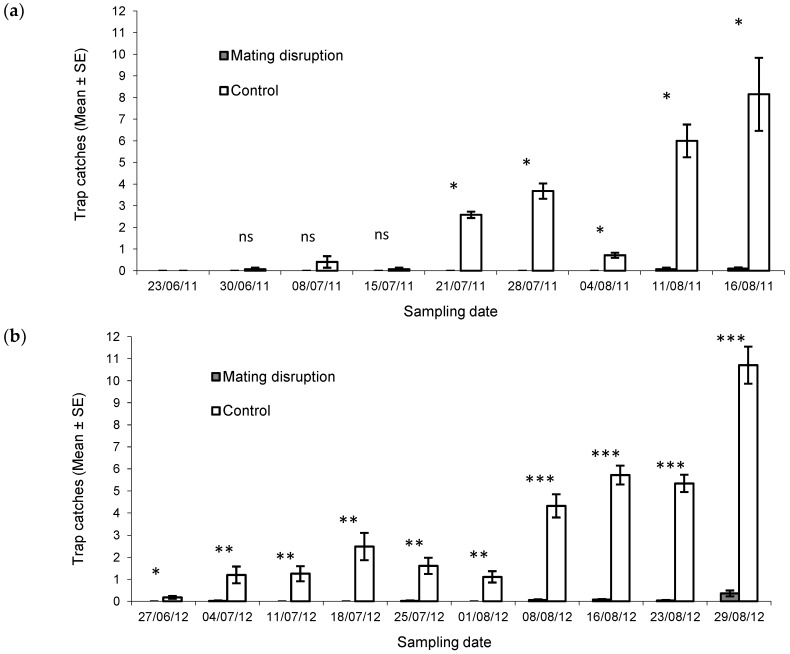
Mean number of *Helicoverpa armigera* males trapped per night in pheromone-treated (mating disruption) and pheromone-untreated (control) areas in 2011 (**a**) and 2012 (**b**). Bars represent the standard errors of the means. Male catches were compared using the Mann–Whitney U test: ns = not significant; * *p* < 0.05; ** *p* < 0.01; *** *p* < 0.001.

**Figure 2 insects-11-00206-f002:**
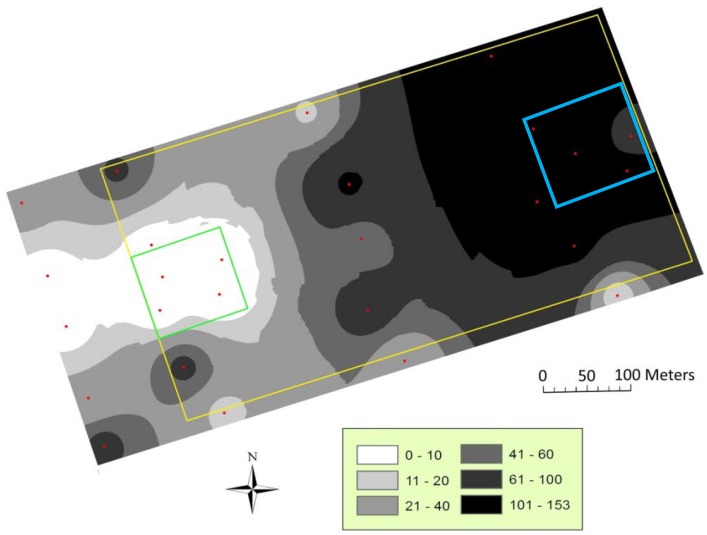
Distribution map of *Helicoverpa armigera* males in 2011 calculated from the total catch per trap during the sampling period (23 June to 16 August). The mating disruption area (1 ha) is represented by the small square on the left; the control area (1 ha) is represented by the small square on the right. Sampling points (pheromone baited traps) are represented by dots.

**Figure 3 insects-11-00206-f003:**
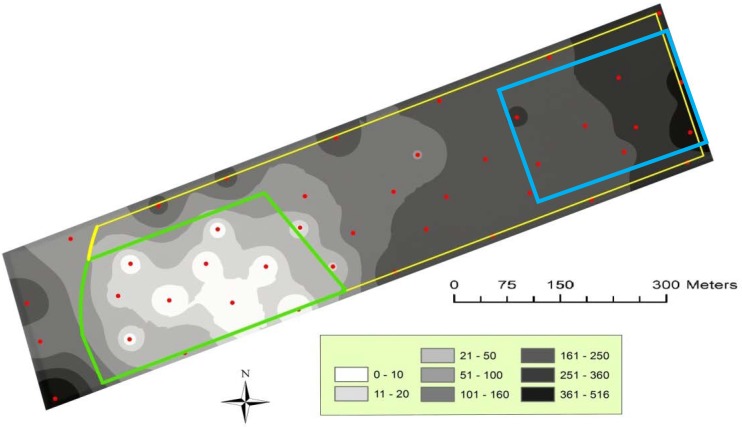
Distribution map of *Helicoverpa armigera* males in 2012 calculated from the total catch per trap during the sampling period (27 June to 29 August). The mating disruption area (5 ha) is represented by the area on the left; the control area is represented by the rectangle on the right. Sampling points (pheromone baited traps) are represented by dots.

**Figure 4 insects-11-00206-f004:**
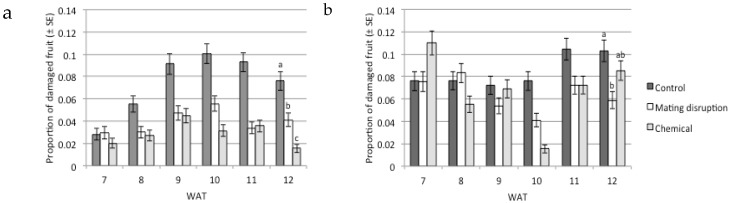
Proportion of damaged fruit (±binomial SE) for the treatments in the 2011 (**a**) and 2012 (**b**) seasons at each sampling date. WAT = weeks after transplanting. WAT 12 corresponded to the last sampling, one day before harvest. Bars bearing different letters are significantly different for *p* < 0.05 (*χ*^2^ test followed by the z-test).

**Table 1 insects-11-00206-t001:** Insecticide sprays applied to tomato fields for *H. armigera* control.

Year	Day	Commercial Name	Active Ingredient	Dose kg c.p./ha
2011	04 July	Steward^®^	Indoxacarb (30 g/L)	0.125
	21 July	Affirm^®^	Emamectin Benzoate (0.95%)	1.5
2012	19 July	Affirm^®^	Emamectin Benzoate (0.95%)	1.5
	10 August	Steward^®^	Indoxacarb (30 g/L)	0.125

c.p., commercial product.

**Table 2 insects-11-00206-t002:** Ratio of catches reduction (%) in the pheromone-treated (mating disruption) area with respect to the pheromone-untreated (control) area in 2011 and 2012.

Year	Weeks after Dispenser Position												
	1	2	3	4	5	6	7	8	9	10	11	12	Total
2011	NA	NA	NA	-	100	100	100	100	100	100	98.8	98.8	99.1
2012	100	98.5	100	100	98.9	100	98.8	98.6	99.3	96.7	NA ^1^	NA ^1^	98.4

NA = not assessed; ^1^ Tomato harvested; - = no male catches.

**Table 3 insects-11-00206-t003:** Results of the cross-validation analysis of the inverse distance weighting (IDW) maps. The linear correlation of the predicted against measured values are reported.

Year	Map	R	*p*	Mean Prediction Error
2011	July, 21	0.80	<0.001	0.01
	July, 28	0.75	<0.001	−0.74
	August, 4	0.24	>0.05	−0.01
	August, 11	0.45	<0.05	−0.44
	August, 16	0.45	<0.05	0.34
	Total catches	0.71	<0.001	−0.91
2012	August, 1	0.42	<0.05	−0.62
	August, 8	0.81	<0.001	−0.99
	August, 16	0.69	<0.001	−2.88
	August, 23	0.57	<0.001	−1.79
	August, 29	0.70	<0.001	−3.67
	Total catches	0.73	<0.001	−13.6

**Table 4 insects-11-00206-t004:** Log linear analysis of fruit damage. Only the interaction of response variable (fruit damage) with design variables (treatments–years–plots–sampling dates) are considered in the analysis.

Effect	*χ* ^2^	*d.f.*	*p*
Treatments * Damage	103.2	2	<0.01
Years * Damage	80.1	1	<0.01
Date * Damage	77.3	5	<0.01
Treatments * Years * Damage	66.7	2	<0.01
Treatments * Date * Damage	48.5	10	<0.01
Years * Date * Damage	39.7	5	<0.01

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
