# Peer review of "Mating Disruption of Helicoverpa armigera (Lepidoptera: Noctuidae) on Processing Tomato: First Applications in Northern Italy"

_insects, 2020, doi:10.3390/insects11040206_

Round 1

Reviewer 1 Report

My previous comments on the not valid nature of this unreplicated design remain problematic.

Reviewer 2 Report

This manuscript reports the efficiency of mating disruption for the cotton bollworms on tomato fields in Italy. According to the authors, the mating disruption method seemed to reduce damage by the cotton bollworms. This result may be valuable to improve the technique of the Integrated Pest Management strategy. However, the method of statistical analysis must be reconsidered.

Major point
The log-linear analysis was used in the section of fruit damage estimation. Years, treatments, sampling dates, and plots were used as independent variables. Then, the authors removed plots as an independent variable after the primary analysis. However, each sampling date is not mutually independent. It is apparent that value today is similar to the value yesterday. The plots are not mutually independent in a similar manner. Because each plot placed near to other plots, it is apparent that each plot shows a similar value. Therefore, these variables should not be added as independent variables. It also is improper to ignore these variables. I recommend the authors to handle the plot ID in a year and sampling dates as a random effect under the generalized linear mixed model.
Although there is a clear difference in the number of males captured between pheromone-treated and untreated areas (Figure 1), the authors also should confirm the statistical analysis of the number of males captured.

Minor points

Inline 205, the sentence lacks the verbs.

In Table 4, the degree of freedom increase with the number of independent variables. This sounds strange to me. I think that the degree of freedom decreases with the number of independent variables.

Reviewer 3 Report

To the authors,

The revision has cleaned up many of the grammatical issues, but there remains several consistency and clarity issues that need to be addressed.

Authors have flushed out the introduction to include a few more references. They do not elaborate on these examples but should. 2.2 Mating disruption trials – would be better to keep the terminology consistent (also in Sections 2.4 and 2.5). The treatments are described as: i) mating disruption, ii) chemical control, iii) untreated control (Lines 75-77). Then on Lines 79-90 the authors make mention of ‘a 1-ha pheromone treated area and a 1-ha pheromone untreated control’….. if these are the mating disruption and chemical control areas, then keep the terminology the same. Or is the ‘1-ha pheromone-untreated control area’ the ‘untreated control’ with the rest of the 18 ha being treated with insecticide and serving as the ‘chemical control’ area? Might be easier to move Line 87 ahead in this paragraph so the reader knows that the main treatment (chemical control) was applied to 18 – 2 ha and that 2 ha were designated as the mating disruption and untreated control areas and received no insecticides. “2.2 Mating disruption trials” (remove the word ‘planning’) Suggest removing “2.3” and “2.4”. These are all part of the ‘mating disruption trial’ section. This is how you are evaluating the impact of the mating disruption technique. No need for a separate numerical heading. ‘Fruit damage estimation’ would then become ‘2.3 Fruit damage estimation’. The in the results, you have 1 section about the mating disruption technique and a second section detailing the fruit damage. Line 130 – Change the header to ‘2.6.1 Mating disruption analysis’ Line 131 – ‘The ratio of catch reduction in the mating disruption area relative to the untreated control area was calculated as follows: [(#male C - #maleMD)/#maleC] x 100; where #maleC = mean number of males caught in the untreated control area and #maleMD = mean number of males caught in the mating disruption area.’ Keep the terminology consistent throughout the manuscript. Lines 136, 137 and 142, Figure 1, Table 2 – same issue. Line 144 – please clarify what your ‘observed values’ were – one would assume ‘trap captures’ but please clarify here. 2.6.2 Fruit damage assessment (not ‘estimation’) – you are describing how you analysed the data here. Authors do not clarify whether the proportion of damaged fruits is on a per plant basis. As there were multiple sampling times, authors also need to clarify if the data were pooled across time, or examined by date individually. Need to state this so the reader can assess that the analysis was properly performed. Line 148 says ‘proportion’ and Line 150 says ‘ratio’. Please keep the terminology consistent. Authors did not address any of the originally stated issues with description of their analysis methods. Line 156 – if ‘plot’ was not significant, please provide the P value here. Line 167 – shouldn’t this be ‘Mating disruption trials’? The Male capture evaluation was a part of the mating disruption so there should be a heading for mating disruption. The male capture was a means to evaluate how well the mating disruption performed. Figure 1 – if the purpose is to only compare the mating disruption with the untreated control area (which is what is shown here) then please remove ‘chemical control’ as a treatment in your methods. Otherwise, there should be 3 bars presented ‘chemical control’, ‘mating disruption’ and ‘untreated control’. Or explain why there are no data for the chemical control – when it was indicated that monitoring was used for the IPM program.

Table 3 – switch ‘year’ and ‘map’ so 2011 and 2012 need appear only once rather than multiple times within the column. Not clear on the benefit of this table.

Year

Map

R

P

Mean prediction error

2011

21 July

0.80

28 July

0.75

4 August

0.24

Table 4 – if ‘Damage’ is the response variable, then it should not be included as an ‘effect’. Here the authors are clearly using 3 treatments, so keep all reported data consistent with this. Remove ‘*Damage’ for all the effects in this table.

Round 2

Reviewer 1 Report

The research is psuedo-replication and does not meet my standards for a valid experimental design

Reviewer 2 Report

The manuscript contains a potentially valuable report of mating disruption for the cotton bollworms.
However, the manuscript has a statistical problem generated by the low number of true replicates.
This study does not distinguish the effect of “place” including each “plot” and the effect of “treatment”.
The true replicate is only “year”.
Two factors, “plot” and “date” cannot be handled as fixed independent variables.
However, this problem does not humble the value of raw data.
I don’t complain if the authors remove the regression analyses and mention to the insufficiency of replicates and if the editor allows the insufficiency of replicates.

As the authors mentioned in the manuscript, the date recorded in the 12th week is the most important.
The authors show that the fruit damage significantly differs between the control and mating disruption groups by Chi-square test.
This pattern observed in both of two years.
Although this replicate (n=2) may be insufficient to conclude that the difference dues to the effect of treatment, this difference seems to be statistically certain.
Therefore, the manuscript may be acceptable If the authors mention the insufficiency of replicates in the discussion because it is true that a significant difference between control and mating disruption groups is obtained twice.
In this case, the regression analyses should be removed.

If the editor can approve the insufficiency of replicates, I recommend the major revision as the decision. If not, I must reject the manuscript in the present circumstance. 

Author Response

This manuscript is a resubmission of an earlier submission. The following is a list of the peer review reports and author responses from that submission.

Round 1

Reviewer 1 Report

This paper presents the results of studies evaluating mating disruption for control of Helicoverpa armigera on tomato in Italy. 

The efficacy of the treatment was evaluated by male captures in pheromone-baited traps, and damage to tomatoes. The manuscript seems to demonstrate that disruption is a viable option for managing this pest in tomato. Unfortunately, I do have a major concern with this paper. In each of the two years of the study the authors treat a single tomato field with pheromone, sampled in another tomato field using IPM to manage the pest, and sampled in a third field that was left untreated. They sample with pheromone traps and fruit injury counts n four areas of each plot. They argue this is some kind of nested design. I argue that the experiment was not replicated in either year. This was pseudo replication. The mating disruption and IPM treatments were only applied to a single plot in each year. They are the treatments that need to be replicated in different tomato fields. And the control needs to be replicated as well. The authors opt to use traps and crop damage in different areas of the plots as replicates. They are not replicates making the statistical analysis of the data invalid. I realize it is hard to replicate mating disruption studies, but this must be done correctly. I reject this paper based on the lack of replication.

If the authors can somehow convince the editor that this was replicated by using multiple traps and multiple damage samples in single plots, then I have two other concerns that should be addressed. The authors state in the abstract and the text that the mating disruption treatment was only partially effective. Those working with mating disruption would consider >95% inhibition of traps and reduction in crop damage as excellent control using this technique. It is understood that the approach is usually density dependent making it difficult to achieve a very high level of disruption. In addition, the authors spend considerable time discussing how the geostatistical analysis of catch in traps reflects the pattern of movement of the pheromone around the plot. Recent studies on mating disruption show that catch in traps may tell us more about pest movement rather than pheromone dispersion

Reviewer 2 Report

This manuscript reports a result of field application of mating disruption method for Helicoverpa armigera, a significant pest insect. Introduction and methods were well written, and results and discussion were simple for readers to understand. The authors clearly showed that mating disruption had higher or similar efficacy than the standard IPM, although both methods can not fully avoid damages on fruit damage of tomatoes. This result is valuable for future aggrecalture that attempt to reduce the amount of chemical insectcides.

As minor points, unnecessary spaces were found at L143 and L181. These spaces should be deleted.

Reviewer 3 Report

Summary

Authors set out to evaluate the ability of mating disruption to reduce damage to organic tomato by cotton bollworm (Helicoverpa armigera). Treatments were intended to compare mating disruption only, chemical sprays only, and no treatment at all.

The experimental design is very poorly described, but here I will try to explain it. Experimental treatments were established each year in a single field and only temporally replicated over a 2-year period (the fields changed between the 2 years). In each year, there were 2 large plots with and without pheromones, both of which were sprayed with insecticides except for within 4 subplots that were insecticide free. Trapping of male moths was carried out in the plots with and without the pheromones (unclear if this was in any of the subplots or not), and then tomato damage was evaluated (apparently in the subplots) with pheromone only, pesticide sprays only and subplots without pheromones or pesticides. Unfortunately, this experimental design is inadequate, as it suffers from an almost complete lack of replication. The subplots within the larger areas with and without pheromone are pseudo-replication. Furthermore, the plots are very small for a mating disruption trial and I’m concern about pheromone drift between the plots, even if upwind.

This manuscript also needs significant revisions by a native English speaker. As mentioned, it is very difficult to understand the methods as currently described.

Reviewer 4 Report

The manuscript evaluates the use of mating disruption for control of cotton bollworm in processing tomato in Italy.  The experimental design is scientifically sound and the methods used to compare treatment areas are reasonable.  The use of geostatistical methods is interesting and provides some nice supportive information to the field results. Authors should improve the manuscript by responding to the numerous suggestions provided on the manuscript directly (see attached) with specific areas needing attention described below.

Introduction needs to be more robust. There are many examples of mating disruption, the authors need to present some examples of successes (preferably in the Noctuidae family) before focussing in on H. armigera. They reference that it is effective but also variable but give no details to describe what could have caused the variability. If the method is effective, then why do it again? Authors need to provide better justification for their study. Also need to be more clear about what a ‘standard IPM strategy’ is – in some areas IPM simply means to monitor first, then apply a conventional insecticide, in other areas it’s a series of cultural practices including companion plantings in addition to pheromone monitoring in the event that a biological application (spray or bio control agent) is required. What is a ‘standard IPM strategy’ in this region? Site description needs more detail as well. How large an area? All of it in processing tomato? Are the geo-coordinates for the area in general, or the pilot farm where the study occurred? Mating disruption trials – not clear which treatment is the ‘IPM method’ referred to in the introduction. Need to be more clear about how the insecticides were applied to the field (Lines 75-77). As written it seems as though the entire area received insecticides, then the pheromone treatment(s) went in afterwards. Need to clarify that these areas were NOT treated with insecticides. Male captures evaluation – need to flush out the details of where the traps were placed, Lines 96-99, and also Lines 102-104. Would be helpful to state what the geo-reference points were used for…. Map development? Would not the addition of this many additional pheromone sources further increase the concentration of pheromone in the areas where these were deployed? Are these close to the mating disruption dispensers, or spaced farther from them? Need to provide justification on the addition of these other traps. Data analysis – male captures. Need to elaborate on what you did with the data. Lines 151-153 describe a ‘suppression ratio’ yet no mention of this occurs in these methods (only the spatial analysis). Fruit damage estimation (Lines 131-148) – was the fruit damage calculated for each plant? Or over all 10 plants in the sampling? If the latter, then the replicates are the sampling dates within each year. The potential issue with sampling date as the replicate is that the damage could be increasing over time. Would suggest the authors look at fruit damage within each sampling on a per plant basis and conduct a preliminary analysis to determine if ‘date’ is significant within each year….. I’m assuming there are several tomatoes on a single plant? Results – reorganization. I think you need to present the results of the mating disruption piece first, then provide the Male capture evaluation and the maps. This makes it consistent with the order presented in the methods. I would also suggest moving Section 2.5 in your methods to above the male captures evaluation (now Section 2.4). Figure 1 – suggest being consistent with treatment names: ‘mating disruption’ also referred to as ‘pheromone-treated’. I would go with ‘mating disruption’ throughout. Use same scale for both panels to facilitate easier comparison of the years. Figures 2 and 3 – where is the ‘control’ area? Would be good to show that as well as the mating disruption area. Table 2 – there’s a dash for 2011 under ‘4’ which is not foot-noted. What does that mean? Lines 189-190 – were the data re-analyzed following discovery of significant interaction effects? It seems to be indicated that this is what was done, yet there are no stats provided (other than P-values in the text). More complete presentation of the statistical results is required. Figure 4 – needs to have a statement about the letters above the bars (and why for only WAT 12) in the caption. Use the same scale for both panels – makes it easier to compare the two years. Table 4 – would suggest adding the basic analysis suggested on lines 186-188 as well. With everything showing significant in the interactions, strongly suggest that authors show statistical results of analysis within each year (as data is shown by year in Figure 4). Reference 16 – Zar. There is a more recent edition of this text.
